# Accuracy of information on the underlying cause of death: An analysis in Colombia during the COVID-19 pandemic in 2021

Pablo Chaparro-Narváez, Jessika Alexandra Manrique Sanchez, Laura Berrio-Parra◉*, Diana Carolina Urrego Ricaurte◉, Luis José Torres-Rojas◉, Nidia Patricia Orjuela Cantor◉, Claudia Patricia Mora Aguirre, Yesid Rojas Quevedo, Clara Suárez, Carlos Castañeda-Orjuela

National Health Observatory, National Institute of Health, Bogotá, Colombia

* lberrio@ins.gov.co, lilio_berrio@hotmail.com

## Abstract

### Objective

This study aimed to estimate the accuracy of the underlying Cause of Death (CoD) in the original death certificate, compared with a gold standard certificate based on information from clinical records and relatives, in population deceased in Colombia during 2021.

### Methods

A sample size of 806 deaths across 92 municipalities in Colombia were estimated from the pool of 326,833 original certificates provided by the National Department of Statistics. A two-stage stratified random sample with replacement was employed for selection. Information from medical records of the deceased and, when necessary, interviews with relatives or witnesses were used to determine CoDs on the gold standard certificate. We analyzed and compared the underlying CoD of the original and standard death certificates to estimate the level of accuracy. Measures of concordance, patterns of false positives and negatives, and a kappa value were utilized as metrics to evaluate the death certificates quality.

### Results

Information was obtained from 776, representing 96% of the desired sample. The concordance between original and gold standard certificates, categorized according to the ICD-10 chapters, was found in 74%. Higher levels of agreement were observed for "codes for special COVID-19 situations" (kappa = 0.84) and neoplasms (kappa = 0.84). Higher levels of agreement were observed for "codes for special COVID-19 situations" (kappa = 0.84) and neoplasms (kappa = 0.84). Overestimation

**Data availability statement:** All relevant data are in the manuscript and its Supporting Information files.

**Funding:** The CDC funded the research, providing financial support for the researchers and to do the field work necessary to collect the information. URL: https://www.cdc.gov/index.htm The CDC did not pay a salary directly to the researchers. The financing was made to the public budget of the National Institute of Health. This study does not have grant numbers per author because it is not a Grant. "The funders had no role in study design, data collection and analysis, decision to publish, or preparation of the manuscript."

**Competing interests:** None of the authors have any conflicts of interest.

was identified for "circulatory system diseases" (Chapter IX); "pregnancy, childbirth and puerperium" (Chapter XV); "signs, symptoms, and poorly defined conditions" (Chapter XVIII) and "diseases of the respiratory system" (Chapter X), while underestimation in "diseases of the genitourinary system" (Chapter XIV) among CoD. The most significant variations in the fraction of mortality due to specific CoDs corresponded to "codes for special situations COVID-19".

## Conclusions

The level of concordance between the original and gold standard death certificates was deemed adequate. However, improvements in the death certification process in Colombia are recommended, emphasizing the enhancement of training programs for health professionals.

## Introduction

The COVID-19 pandemic highlighted significant vulnerabilities within global Civil Registration and Vital Statistics (CRVS) systems, underscoring the urgent need to enhance these information frameworks [1]. Effective public policy formulation is contingent upon the availability of comprehensive, detailed, continuous, and real-time statistics related to population health and mortality -key outputs of a robust CRVS system [2]. Encompassing vital events such as births, deaths, and others, the CRVS system not only offers legal and administrative benefits to individuals and families but also provides critical statistical and administrative data for policymakers. The continuous registration of deaths, along with their causes, is fundamental for informed policymaking aimed at reducing premature mortality. In systems with inherent weaknesses, responses to health threats may be delayed, leading to a higher incidence of deaths. Conversely, timely mortality statistics enable ongoing surveillance, reflecting trends in death burden from various causes, and furnishing essential up-to-date data for informed decision-making [3].

Information on the Cause of Death (CoD) plays a pivotal role in guiding decision-making process, influencing policy direction, research funding priorities, and evaluating the effectiveness of public health interventions [4,5]. Seven Sustainable Development Goals (SDGs) specifically rely on mortality data sourced from CRVS systems [6]. However, the accuracy and timeliness of data recorded on death certificates can exhibit significant variability [7–9]. Deficiencies in CoD information quality can stem from factors such as inadequate training among coders, unfamiliarity with causes [7,10]., time constraints [6], and a lack of supervision, leading to the frequent use of imprecise "garbage codes" [11]. Additionally, the implementation of effective procedures to consolidate information and adherence to statistical standards that transform individual codes into comprehensive national mortality statistics are crucial [8]. Notably, errors in CoD reporting are prevalent, with rates ranging from 39% to 61% in population-based studies and 32% to 45% in hospital studies [7–9].

In 2019, the reporting landscape in the Americas showed an estimated underreporting rate of 4%, with poorly defined or absent CoD codes at 3%, and garbage codes at 15%. in contrast, Colombia experienced higher underreporting at 13%, with poorly defined and ignored CoD at 1% and garbage codes at 11% [12].. Despite these statistics, studies on CoD in Colombia have been limited. Existing research has primarily focused on various aspects such as the knowledge of different CRVS actors [13], the overall quality of certification in general and cancer mortality [14–16], the certification quality of live births and non-fetal deaths in hospitals [15,17]., and the quality of information for children under one year of age [18]

In Colombia, the National Statistics Institute (DANE by its Spanish acronym) is the governmental entity responsible for official statistics, including vital statistics. In the 1990s, the Civil Registry and Vital Statistics System (SRCEV by its Spanish acronym) was established to organize and standardize information on births and deaths. This system comprises the Civil Registry and Vital Statistics subsystems [19]. The Civil Registry subsystem registers and stores information about vital events and their characteristics for legal and administrative purposes [20]. The Vital Statistics subsystem collects, processes, and disseminates information on all births and deaths that occur in the country, enabling DANE to prepare official statistics of these events [19].

In 2019, the "National yearbook of vital statistics Colombia 2019" was published, dedicating a chapter to data quality, but excluding the quality of the underlying CoD information [21]. Studies evaluating basic CoD selected according with the International Classification of Diseases, 10th Revision (ICD-10) coding rules, comparing certificates prepared by a certifier (original certification) with those by researchers (gold standard certification), are scarce [17–18]. Shortcomings in both completeness and accuracy pose significant challenges to accurately accounting for deaths. Inaccurate identification of CoD can detrimentally affect the reliability of public health datasets, impeding both local and national response strategies. Our study aims to address this research gap by estimating the accuracy of underlying CoD in original death certificates compared with gold standard certificates based on information from clinical records and relatives for individuals who died in Colombia during 2021.

## Methods

### Study design and data collection

This cross-sectional descriptive study was conducted in Colombia in 2021. The death certificate in Colombia is designed according to international standards, especially in aspects related to the CoD. Doctors usually fill out the death certificate. DANE is responsible for coding and selecting the underlying CoD following ICD-10 standards. In 2022, DANE used the 2019 version of ICD-10 and employed the automated coding system IRIS. For records not coded by IRIS, manual assignment of the ICD-10 code was performed [22].

### Sampling

Death data were obtained from individual death certificates consolidated in DANE's mortality databases. Initially, DANE had compiled 326,833 non-fetal death certificates, excluding those related to violent causes certified by the Institute of Legal Medicine and Forensic Sciences, as well as municipalities without reported COVID-19 infections or deaths. The study focused on municipalities classified as large and intermediate, where COVID-19 cases were identified based on healthcare service complexity and the presence of more than 20 deaths associated with COVID-19, as determined from medical records or death certificate.

A two-stage stratified random sample with replacement was designed. In the first stage, it was considered conglomerates comprising 33 territorial entities of country's administrative segmentation (32 departments and the district of Bogotá). In the second stage, from 770 municipalities, were 92 selected using random sampling with computer-generated random numbers, stratified by age groups (<60 years and ≥60 years). The estimated sample size was 806 deaths across the 92 municipalities, with a confidence level of 95%, an expected proportion of 70%, an estimation precision of 5%, and an effect of 2.5 (due to heterogeneity on the municipal mortality concentration). The minimum expected ratio is p, this varies

according to the health system, local practices and the method used. It is reasonable to assume that, under appropriate conditions of quality and training in medical certification, the 70% ratio can be considered representative and feasible in systems that prioritize accuracy and consistency in mortality records [23–26]. The program EPIDAT 4.2 was used for the random selection of the sample [27].

## Data collection and analysis

Health professionals, including five doctors and a nurse with a postgraduate degree in epidemiology and public health collected and analyzed information from medical records. These professionals underwent prior training in the DANE's vital statistics to ensure accurate CoDs assignment in the gold standard certificate. Medical records reviews took place between June 1, 2022, and September 30, 2022.

Identification data of the deceased from the original death certificate were used to request corresponding medical records from territorial health entities. Under strict supervision, health professionals extracted information to complete the underlying CoD in the gold standard certificate. To mitigate bias, these professionals did not have access to the original death certificate. In cases where medical records lacked sufficient information, interviews with relatives, acquaintances, or witnesses were conducted. Health professionals conducted telephone interviews with relatives or friends of the deceased in cases where the clinical history was not available, or the information recorded was insufficient to determine the CoD. During these interviews, they were asked about the events that occurred during the illness or the situation that led to the person's death.

An expert nosologist coded the CoD assigned by the health personnel, selecting the underlying CoD based on ICD-10 norms used in Colombia`s official mortality statistics. For the study period, the 2019 version of ICD-10 was used [28]. There was no CoD adjudicating committee; instead, the CoDs were directly collected from the medical records by the research team. This gold standard certificate, based on detailed medical records, was deemed the most accurate source.

## Statistical analysis

The underlying CoDs in the original (prepared by the certifiers) and gold standard (prepared by the researchers) certificates were compared at the ICD-10-chapter level and mortality tabulation list 2–80 causes. The overall concordance rate between the underlying CoD in both certificates, the percentage of positive concordance (number of cases in which both data sets report a positive result), false positive rate, and false negative rate were calculated. Additionally, the kappa coefficient was computed to measure the degree of agreement beyond chance [29–32]. The 95% confidence intervals (95% CI) for these metrics were also estimated. The formulas used for these calculations are as follows (Table 1):

$$\text{Overall Concordance Rate} \ = \ (a+d)/(a+b+c+d)$$

$$\text{False Positive Rate} \ = \ b/(a+b)$$

$$\text{False Negative Rate} \ = \ c/(c+d)$$

**Table 1. Methods of calculating different indicators of agreement.**

| | | Reviewer underlying cause | |
|---|---|---|---|
| | | Diagnosis X | Diagnosis X |
| Original underlying cause | Diagnosis X | a | B |
| | The others | c | D |

$$\text{Kappa Coefficient} = (Po - Pe)/(1 - Pe)$$

Where:

$$Po\ (\text{Observed percentage of agreement}) = (a + d)/(a + b + c + d)$$

$$Pe\ (\text{expected percentage of agreement}) = ([a+c]/[a+b+c+d]) * ([a+b]/[a+b+c+d]) + ([b+d]/[a+b+\ c+d]$$

False positive and false negative percentages were used to quantify the degree of overdiagnosis and under-certification, respectively [33]. The kappa statistic is a widely used measure for evaluating validity and reliability [32], accounting for occurring by chance between observers of the underlying CoD in the gold standard and the original certificates [33]. Kappa values greater than 0.81 were considered almost perfect, 0.61 to 0.80 as high, 0.41 to 0.60 as moderate, 0.21 to 0.40 as acceptable, 0.00 to 0.20 as low, and those < 0.00 as poor [30]. The Change in Cause-Specific Mortality Fraction (CCSMF) was calculated to assess the impact of discrepancies between the original and gold standard certificates [34].

$$CCSMF = \frac{CaSMF^S - CaSMF^O}{CaSMF^O} \times 100$$

Where *CaSMF* = cause specific mortality fraction, O = "original" certificate, S = "standard" certificate.

$$CaSMF = \frac{Number\ of\ deaths\ due\ to\ one\ specific\ cause}{Total\ number\ of\ deaths}$$

To further assess variations in the underlying CoDs between original and gold standard certificates, the Bland-Altman diagram was employed. This visual tool helps in identifying any systematic differences between the two sets of data. The collected information was processed using Microsoft Excel spreadsheets and analyzed using Stata, version 12.0.

### Ethical considerations

The project received approval from the Committee on Ethics and Research Methodologies of the National Institute of Health (CEMIN, by Spanish acronym). Anonymized mortality databases from DANE were accessed, and health professionals extracted information from medical records without reviewing the original death certificates to prevent bias. CEMIN authorized the review of medical records without Informed consent, considering the National Institute of Health's role as the national health authority. These documents were securely held by research team and the respective territorial entities.

To address gaps in the medical records, researchers contacted the residences of the deceased, aiming to speak with a relative or acquaintance. Researchers identified themselves as representatives of the National Institutes of Health, explained the study's purpose, and outlined the protocol. Consent was obtained from participants for their involvement in the study and for the audio recording of the conversation. If participants declined, they were thanked, and a follow-up call was made. If audio recording was not permitted, permission was sought to take written notes.

During the interviews, specific questions regarding gaps in the deceased's medical history were asked. Participants were informed that they could decline to answer any question without providing a reason. Ensuring ethical standards and maintaining participant confidentiality throughout the research process was paramount, necessitating rigorous anonymization of the data. Interview notes and audio recordings were stored in the INS institutional cloud, password-protected, and accessible only to project researchers. This process was endorsed by CEMIN, ensuring ethical standards and the protection of participants' confidentiality throughout the research process.

## Results

In our sample of 776 deaths, stratification was conducted by department, municipality, and age, ensuring a representative cross-section of Colombia`s total mortality. Despite a nominal 4% loss in the sample, this minimal reduction did not introduce any distortion to the study's findings. The mean age of death was 65 years (range: 0–98 years; median: 68 years). Most deaths occurred in urban areas (94.2%) and hospitals (78.5%). The determination of CoD primarily relied on medical records (93.9%), with 1% of the sample drawing from a multifaceted array of sources such as medical records, laboratory tests, and interviews with relatives or witnesses (Table 2). Overall, the agreement on the underlying CoD, based on ICD-10 chapter, was robust at 74% (575 deaths), whereas the mortality rate based on ICD-10 mortality tabulation list 2 was 63.4% (492 deaths correct classified) (S1 Table).

### Stratification and agreement analysis

The stratification of the underlying CoD between original and gold standard death certificates is noteworthy (Table 3). In 18 instances where the gold standard certificate identified "COVID-19" as the underlying CoD, the original certificate asserted "diseases of the respiratory system" (Chapter X). Conversely, in six cases where the gold standard certificate

**Table 2. Characteristics of analyzed deaths. Colombia, 2021.**

| | Women (n = 352) | | Men (n = 424) | | Total (n = 776) | |
|---|---|---|---|---|---|---|
| | n | % | n | % | n | % |
| Age | | | | | | |
| Mean age at death (years) | 65.9 (SD: 21.4) | | 64.4 (SD: 20.9) | | 65.1 (SD: 21.1) | |
| 0–9 | 12 | 3.4 | 15 | 3.5 | 27 | 3.5 |
| 10–19 | 2 | 0.6 | 6 | 1,4 | 8 | 1.0 |
| 20–29 | 10 | 2.8 | 6 | 1.4 | 16 | 2.1 |
| 30–39 | 15 | 4.3 | 21 | 5.0 | 36 | 4.6 |
| 40–49 | 22 | 6.3 | 31 | 7.3 | 53 | 6.8 |
| 50–59 | 42 | 11.9 | 62 | 14.6 | 104 | 13.4 |
| 60–69 | 80 | 22.7 | 83 | 19.6 | 163 | 21.0 |
| 70–79 | 68 | 19.3 | 97 | 22.9 | 165 | 21.3 |
| 80+ | 101 | 28.7 | 103 | 24.3 | 204 | 26.3 |
| Area of death | | | | | | |
| Urban | 334 | 43.0 | 397 | 51.2 | 731 | 94.2 |
| Rural | 18 | 2.3 | 27 | 3.5 | 27 | 5.8 |
| Place of death | | | | | | |
| Hospital/clinic | 268 | 34.5 | 341 | 43.9 | 609 | 78.5 |
| Health center/post | 1 | 0.1 | 1 | 0.1 | 2 | 0.3 |
| Home | 79 | 10.2 | 75 | 9.7 | 154 | 19.8 |
| Workplace | 0 | 0.0 | 3 | 0.4 | 3 | 0.4 |
| Another place | 4 | 0.5 | 4 | 0.5 | 8 | 1.0 |
| Determination of CoD | | | | | | |
| Necropsy | 0 | 0.0 | 3 | 0.4 | 3 | 0.4 |
| Clinical record | 337 | 43.4 | 392 | 50.5 | 729 | 93.9 |
| Interview to relatives or witnesses | 32 | 4.1 | 40 | 5.2 | 72 | 9.3 |
| Multisource | 6 | 0.8 | 1 | 0.1 | 7 | 0.9 |

SD: standard deviation.

Table 3. Cross tabulation of underlying CoD (ICD-10) in original and gold standard death certificates, Colombia, 2021.

| | ICD-10 chapter | Original death certificated | | | | | | | | | | | | | | | | | | |
|---|---|---|---|---|---|---|---|---|---|---|---|---|---|---|---|---|---|---|---|---|
| | | I | II | III | IV | V | VI | IX | X | XI | XII | XIII | XIV | XV | XVI | XVII | XVIII | XX | XXII | Total |
| Standard death certificate | I | 15 | 1 | | | | | 2 | | 2 | | | | | | | 2 | | 1 | 23 |
| | II | 1 | 109 | | | | 1 | 10 | | 3 | 1 | | 1 | | | | | 1 | 1 | 128 |
| | III | | | 5 | | | | 2 | | | | | | | | | | | | 7 |
| | IV | 1 | | | 17 | | | 5 | | | | | | | | | | 2 | | 25 |
| | V | | | | | 2 | | | | | | | 1 | | | | | | | 3 |
| | VI | | | | | 1 | 13 | 3 | | | | | | | | | | 2 | | 19 |
| | IX | | 4 | 1 | 8 | | 3 | 130 | 6 | 5 | | 1 | 4 | 1 | | 1 | 3 | | 5 | 172 |
| | X | 1 | 3 | 1 | 3 | | 1 | 8 | 26 | 1 | | | | 1 | | 1 | | | 6 | 52 |
| | XI | 1 | 1 | | 2 | | | 5 | 2 | 24 | | | | | | | | | | 35 |
| | XII | | | | | | | | 1 | | 1 | | | | | | 1 | | | 3 |
| | XIII | | | | | | | 1 | | | | | 3 | | | | | | 1 | 5 |
| | XIV | 1 | 2 | | 5 | | | 3 | 1 | 1 | | 1 | 5 | | | | 1 | | | 20 |
| | XV | | | | | | | | | | | | | 0 | | | | | | 0 |
| | XVI | 2 | | | | 1 | | | | | | | | | 6 | 1 | | | | 10 |
| | XVII | | | | | | | 2 | | | | | | | 1 | 6 | | | | 9 |
| | XVIII | | | | 1 | | | 3 | 1 | | | | | | | | 0 | | | 5 |
| | XX | | 1 | | | | 1 | 6 | 2 | 1 | | | | | | | | 1 | 1 | 13 |
| | XXII | 2 | 2 | | 2 | | | 7 | 18 | 1 | | 1 | 2 | | | | | | 212 | 247 |
| | Total | 24 | 123 | 7 | 38 | 4 | 19 | 185 | 57 | 40 | 2 | 6 | 13 | 2 | 7 | 9 | 7 | 2 | 231 | 776 |

I: Infections; II: Neoplasms; III: Blood diseases; IV: Endocrine diseases; V: Mental disorders; VI: Nervous system diseases; IX: Circulatory diseases; X: Respiratory diseases; XI: Digestive diseases; XII: Skin diseases; XIII: Osteomuscular tissue and connective tissue diseases; XIV: Genitourinary diseases; XV: Pregnancy, childbirth and puerperium; XVI: Perinatal conditions; XVII: Congenital anomalies; XVIII: Signs, symptoms and poorly defined conditions; XX: External causes; XXII: COVID-19.

pinpointed "respiratory system diseases" (Chapter X), the original certificate indicated "COVID-19" (Chapter XXII). Notable discrepancies were also observed in eight cases where the gold standard certificate specified "circulatory system diseases" (Chapter IX) as the underlying CoD, while the original certificate pointed to "endocrine diseases" (Chapter IV). Additionally, ten cases with "neoplasms" (Chapter II) as the gold standard certificate's CoD, the original certificate classified them as "circulatory system diseases" (Chapter IX).

## Causes of death

Most CoDs reporting "neoplasms" (Chapter II) or "codes for special situations (COVID-19)" (Chapter XXII) exhibited high consistency between the two sources (Table 4). However, conditions such as "diseases of the genitourinary system" (Chapter XIV) showed elevated rates of false positives (Kappa 0.29; 95% CI: 0.08–0.50), while "diseases of the circulatory system"(Chapter IX) (Kappa 0.65; 95% CI: 0.58–0.71), "codes for special situations (COVID-19)" (Chapter XXII) (Kappa 0.84; 95% CI: 0.79–0.88), and "diseases of the respiratory system"(Chapter X) (Kappa 0.44; 95% CI: 0.32–0.56) displayed substantial false negatives (Table 3). The greatest variations were for "codes for special situations (COVID-19)" (chapter XXII) (6.9%), "diseases of the circulatory system" (Chapter IX) (-7.0%) and "endocrine diseases, nutritional and metabolic" (Chapter IV) (-34.2%).

The proportion of agreement for most CoDs remained consistent between genders, in men was 73% (311 deaths) and in women was 75% (264 deaths). Notably, "neoplasms" (Chapter II) (Kappa 0.82; 95% CI: 0.73–0.89 and Kappa 0.87; 95% CI: 0.80–0.94, respectively) and "codes for special situations (COVID-19)" (Chapter XXII) (Kappa 0.83; 95%

**Table 4. Underlying CoD (ICD-10) agreement metrics between original and gold standard death certificates. Colombia, 2021.**

| ICD-10 chapter | Death numbers | | Match between certificates | False positive % | False negative % | Kappa | SE | Kappa CI 95% | | Change in Cause-Specific Mortality Fraction (%) |
|---|---|---|---|---|---|---|---|---|---|---|
| | Original (O) | Standard (S) | | | | | | LL | UL | |
| I | 24 | 23 | 15 | 37.5 | 1.1 | 0.63 | 0.084 | 0.46 | 0.79 | -4.2 |
| II | 123 | 128 | 109 | 11.4 | 2.9 | 0.84 | 0.027 | 0.79 | 0.90 | 4.1 |
| III | 7 | 7 | 5 | 28.6 | 0.3 | 0.71 | 0.138 | 0.44 | 0.98 | 0.0 |
| IV | 38 | 25 | 17 | 55.3 | 1.1 | 0.52 | 0.078 | 0.37 | 0.67 | -34.2 |
| V | 4 | 3 | 2 | 50.0 | 0.1 | 0.57 | | 0.13 | 1.00 | -25.0 |
| VI | 19 | 19 | 13 | 31.6 | 0.8 | 0.68 | 0.088 | 0.50 | 0.85 | 0.0 |
| IX | 185 | 172 | 130 | 29.7 | 7.1 | 0.65 | 0.033 | 0.58 | 0.71 | -7.0 |
| X | 57 | 52 | 26 | 54.4 | 3.6 | 0.44 | 0.062 | 0.32 | 0.56 | -8.8 |
| XI | 40 | 35 | 24 | 40.0 | 1.5 | 0.62 | 0.067 | 0.49 | 0.75 | -12.5 |
| XII | 2 | 3 | 1 | 50.0 | 0.3 | 0.40 | 0.278 | 0.00 | 0.94 | 50.0 |
| XIII | 6 | 5 | 3 | 50.0 | 0.3 | 0.54 | 0.182 | 0.19 | 0.90 | -16.7 |
| XIV | 13 | 20 | 5 | 61.5 | 2.0 | 0.29 | 0.105 | 0.08 | 0.50 | 53.8 |
| XV | 2 | 0 | 0 | 100.0 | 0.0 | 0.00 | 0.000 | 0.00 | 0.00 | -100.0 |
| XVI | 7 | 10 | 6 | 14.3 | 0.5 | 0.70 | 0.127 | 0.45 | 0.95 | 42.9 |
| XVII | 9 | 9 | 6 | 33.3 | 0.4 | 0.66 | 0.129 | 0.41 | 0.92 | 0.0 |
| XVIII | 7 | 5 | 0 | 100.0 | 0.7 | -0.01 | 0.002 | -0.01 | 0.00 | -28.6 |
| XX | 2 | 13 | 1 | 50.0 | 1.6 | 0.13 | 0.119 | 0.00 | 0.36 | 550.0 |
| XXII | 231 | 247 | 212 | 8.2 | 6.4 | 0.84 | 0.021 | 0.79 | 0.88 | 6.9 |

SE: standard error LL: lower limit UL: upper limit.

I: Infections; II: Neoplasms; III: Blood diseases; IV: Endocrine diseases; V: Mental disorders; VI: Nervous system diseases; IX: Circulatory diseases; X: Respiratory diseases; XI: Digestive diseases; XII: Skin diseases; XIII: Osteomuscular tissue and connective tissue diseases; XIV: Genitourinary diseases; XV: Pregnancy, childbirth and puerperium; XVI: Perinatal conditions; XVII: Congenital anomalies; XVIII: Signs, symptoms and poorly defined conditions; XX: External causes; XXII: COVID-19.

CI: 0.77–0.88 and Kappa 0.84; 95% CI: 0.77–0.91, respectively), demonstrated almost perfect concordance in both genders. However, "circulatory system diseases" (Chapter IX) exhibited higher percentages of false positive (Kappa 0.60; 95% CI: 0.49–0.69 and Kappa 0.68; 95% CI: 0.60–0.76, respectively) (Table 5).

In terms of CSMF attributed to specific CoD, notable variations were identified in both men and women, with the most significant discrepancies observed in "codes for special situations (COVID-19)" (Chapter XXII). Moreover, noteworthy shifts were noted among men, specifically in "endocrine diseases" (Chapter IV) with an increase of +2.2%, and "respiratory system diseases" (Chapter X) also exhibiting a rise of +2.2%. Conversely, women demonstrated substantial variations, particularly in "circulatory system diseases" (Chapter IX) where there was a marked increase of +5.0%.

The Bland-Altman diagram, illustration the absolute difference in underlying CoD between the original and gold standard death certificates (Fig 1), indicates a mean difference was 0 (with limits of agreement: 3.5 to -3.5). The variable plotted in the Bland-Altman diagram was the comparison of underlying CoD grouped by ICD-10 chapter from both the original and gold standard death certificates.

## Discussion

This study explores the complexities of the death certification process in Colombia, offering valuable insights into the accuracy of Cause of Death (CoD) reporting. It was conducted during the COVID-19 pandemic, which could have affected the accuracy of death certification due to the high volume of deaths and the strain on healthcare systems. By examining

**Table 5. Underlying CoD (ICD-10) agreement metrics between original and gold standard death certificates by sex. Colombia, 2021.**

| ICD-10 chapter | Men | | | | | | | | | Women | | | | | | | | | |
|---|---|---|---|---|---|---|---|---|---|---|---|---|---|---|---|---|---|---|---|
| | Death numbers Original (O) | Standard (S) | Match between certificates | False positive % | False negative % | Kappa | Kappa CI95% LL | LS | Change in Cause-Specific Mortality Fraction (%) | Death numbers Original (O) | Standard (S) | Match between certificates | False positive % | False negative % | Kappa | Kappa CI95% LL | LS | Change in Cause-Specific Mortality Fraction (%) |
| I | 13 | 10 | 8 | 1.21 | 20.00 | 0.69 | 0.47 | 0.91 | -23.1 | 11 | 13 | 7 | 1.18 | 46.15 | 0.57 | 0.33 | 0.81 | 18.2 |
| II | 64 | 64 | 54 | 2.78 | 15.63 | 0.82 | 0.73 | 0.89 | 0.0 | 59 | 64 | 55 | 1.39 | 14.06 | 0.87 | 0.80 | 0.94 | 8.5 |
| III | 3 | 3 | 2 | 0.24 | 33.33 | 0.66 | 0.23 | 1.00 | 0.0 | 4 | 4 | 3 | 0.29 | 25.00 | 0.75 | 0.41 | 1.00 | 0.00 |
| IV | 18 | 8 | 5 | 3.13 | 37.50 | 0.37 | 0.13 | 0.61 | -55.6 | 20 | 17 | 12 | 2.39 | 29.41 | 0.63 | 0.44 | 0.82 | -15.0 |
| V | 1 | 0 | 0 | 0.24 | – | 0.00 | 0.00 | 0.00 | -100.0 | 3 | 3 | 2 | 0.29 | 33.33 | 0.66 | 0.23 | 1.00 | 0.0 |
| VI | 10 | 10 | 6 | 0.97 | 40.00 | 0.59 | 0.33 | 0.85 | 0.0 | 9 | 9 | 7 | 0.58 | 22.22 | 0.77 | 0.56 | 0.99 | 0.0 |
| IX | 72 | 78 | 50 | 6.36 | 35.90 | 0.60 | 0.49 | 0.69 | 8.3 | 113 | 94 | 80 | 12.79 | 14.89 | 0.68 | 0.60 | 0.76 | -16.8 |
| X | 37 | 28 | 16 | 5.30 | 42.86 | 0.45 | 0.29 | 0.61 | -24.3 | 20 | 20 | 10 | 3.01 | 50.00 | 0.47 | 0.27 | 0.68 | 0.0 |
| XI | 20 | 18 | 13 | 1.72 | 27.78 | 0.67 | 0.49 | 0.84 | -10.0 | 20 | 17 | 11 | 2.69 | 35.29 | 0.57 | 0.37 | 0.77 | -15.0 |
| XII | 1 | 1 | 0 | 0.24 | 100.00 | 0.00 | -0.01 | 0.00 | 0.0 | 1 | 2 | 1 | 0.00 | 50.00 | 0.67 | 0.05 | 1.00 | 100.0 |
| XIII | 2 | 1 | 1 | 0.24 | 0.00 | 0.67 | 0.05 | 1.00 | -50.0 | 4 | 4 | 2 | 0.57 | 50.00 | 0.49 | 0.07 | 0.92 | 0.0 |
| XIV | 8 | 14 | 3 | 1.22 | 78.57 | 0.25 | 0.01 | 0.50 | 75.0 | 5 | 6 | 2 | 0.87 | 66.67 | 0.35 | -0.01 | 0.72 | 20.0 |
| XV | 0 | 0 | 0 | 0.00 | – | – | – | – | – | 2 | 0 | 0 | 0.57 | – | 0.00 | – | – | -100.0 |
| XVI | 5 | 7 | 4 | 0.24 | 42.86 | 0.66 | 0.35 | 0.97 | 40.0 | 2 | 3 | 2 | 0.00 | 33.33 | 0.80 | 0.41 | 1.00 | 50.0 |
| XVII | 6 | 6 | 4 | 0.48 | 33.33 | 0.66 | 0.35 | 0.97 | 0.0 | 3 | 3 | 2 | 0.29 | 33.33 | 0.66 | 0.23 | 1.00 | 0.0 |
| XVIII | 5 | 3 | 0 | 1.19 | 100.00 | -0.01 | -0.01 | 0.00 | -40.0 | 2 | 2 | 0 | 0.57 | 100.00 | -0.01 | -0.01 | 0.00 | 0.0 |
| XX | 2 | 8 | 1 | 0.24 | 87.50 | 0.19 | -0.14 | 0.52 | 300.0 | 0 | 5 | 0 | 0.00 | 100.00 | 0.00 | 0.00 | 0.00 | – |
| XXII | 157 | 165 | 144 | 5.02 | 12.73 | 0.83 | 0.77 | 0.88 | 5.1 | 74 | 82 | 68 | 2.22 | 17.07 | 0.84 | 0.77 | 0.91 | 10.8 |

I: Infections; II: Neoplasms; III: Blood diseases; IV: Endocrine diseases; V: Mental disorders; VI: Nervous system diseases; IX: Circulatory diseases; X: Respiratory diseases; XI: Digestive diseases; XII: Skin diseases; XIII: Osteomuscular tissue and connective tissue diseases; XIV: Genitourinary diseases; XV: Pregnancy, childbirth and puerperium; XVI: Perinatal conditions; XVII: Congenital anomalies; XVIII: Signs, symptoms and poorly defined conditions; XX: External causes; XXII: COVID-19.

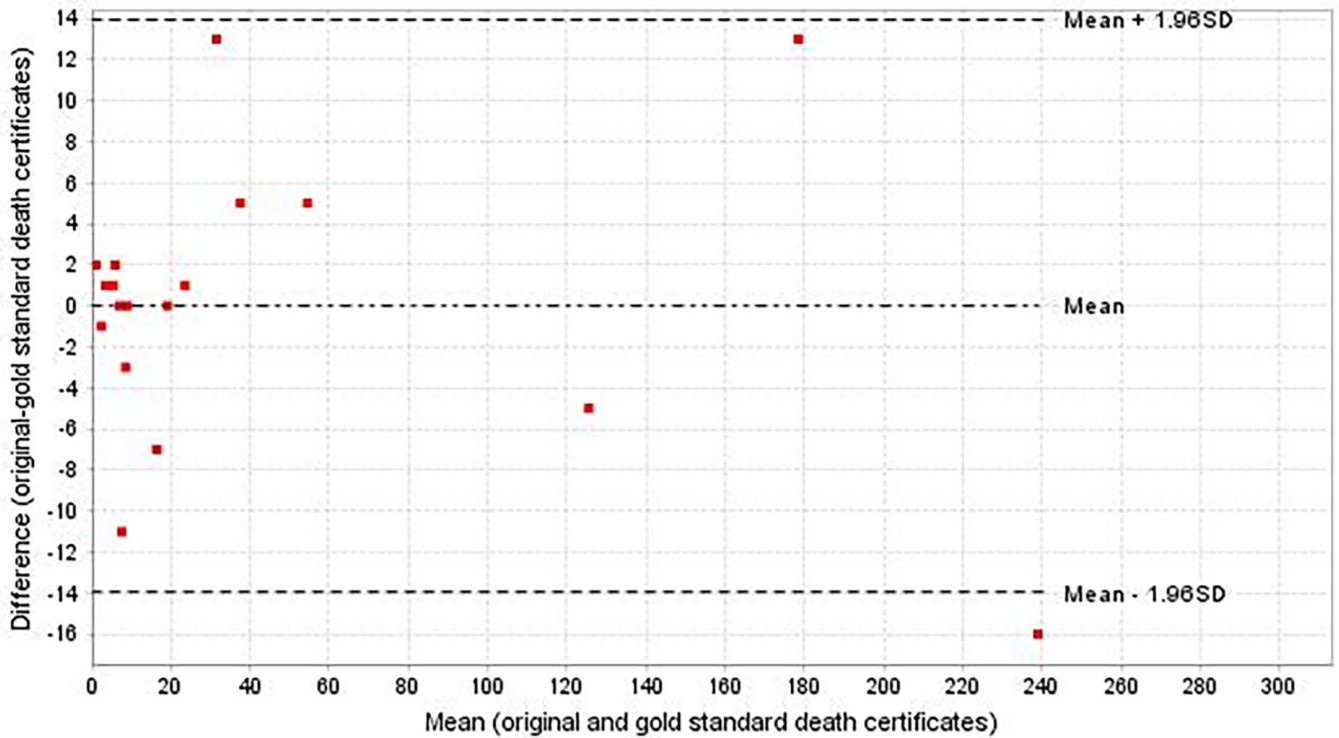

PLOS One logo

a substantial sample size of 776 randomly selected death certificates from 2021, the second year of the COVID-19 pandemic. Our research employed a meticulous retrospective review, drawing upon information documented in medical records. The collaboration with health professionals adept in death certification preparation and CoD assignment added depth to our analysis. The overall agreement between both the original and gold standard death certificates for the underlying CoD information, categorized according to the ICD-10 chapter, stood at a commendable 74%. Further stratification by gender revealed similar accordance in women (75%) compared to men (73%). These findings underscore the nuance landscape of CoD estimation in Colombia, where both overestimation and underestimation contribute to the complexity of mortality statistics.

Comparing our results globally, the accuracy of the underlying CoD in our study surpassed figures reported in Thailand (37%) [35] and Bangladesh (54%) [36], signifying a comparatively higher precision in Colombia. However, it fell short of the 81% observed in Valparaíso and Metropolitan region in Chile [37]. Disparities in age demographics and the unique challenges posed by the ongoing pandemic caution against direct comparisons, emphasizing the need for context-specific assessments.

Our study highlighted notable overestimations and underestimations in specific CoD categories. High concordance was evident for "neoplasms" (Chapter II) and "codes for special situations (COVID-19)" (Chapter XXII), which collectively constituted 33% of deaths in Colombia during 2021. Overestimation (a high percentages of false negatives and negative values in the CSMF) was identified for "circulatory system diseases" (Chapter IX); "pregnancy, childbirth and puerperium" (Chapter XV); "signs, symptoms, and poorly defined conditions" (Chapter XVIII) and "diseases of the respiratory system" (Chapter X), while underestimation (a high percentages of CSMF) in "diseases of the genitourinary system" (Chapter XIV) among CoD.

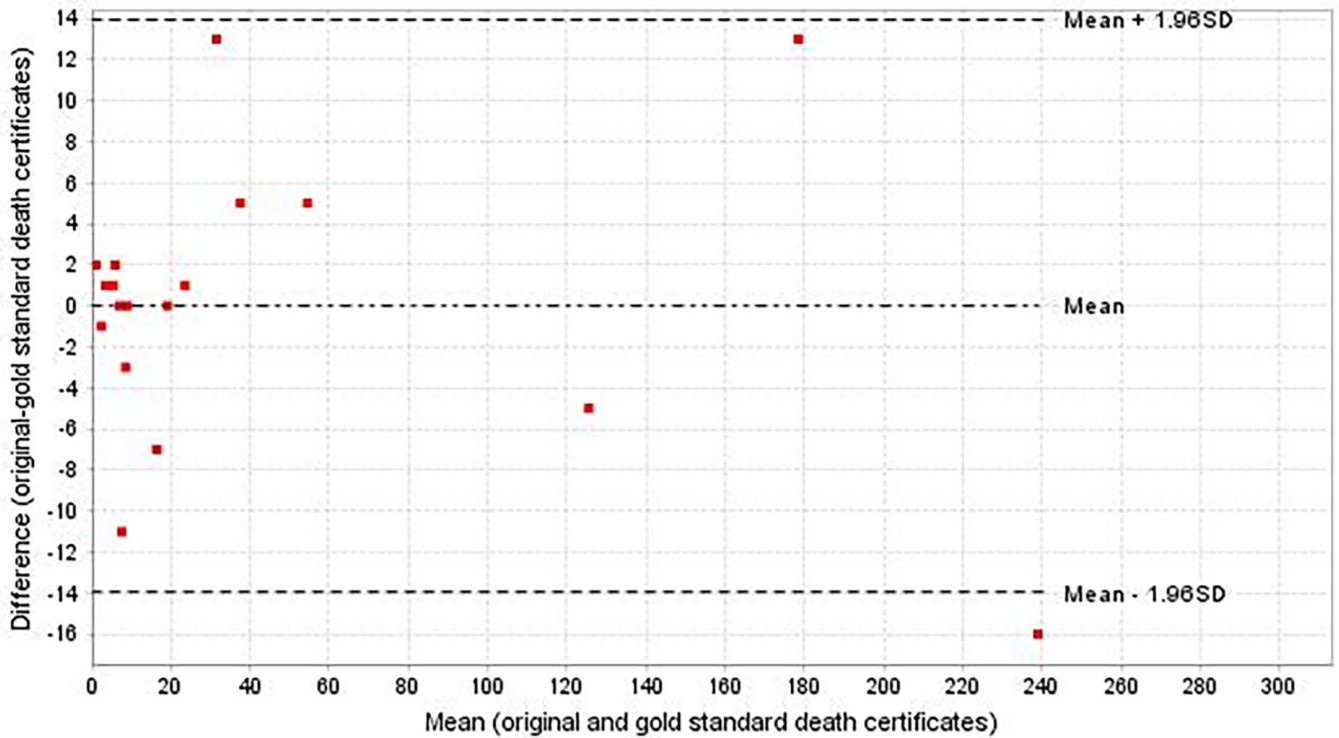

**Fig 1. Bland-Altman plot. Difference between original and gold standard death certificates for underlying CoD.**

The high level of concordance for deaths related to neoplasms, evidenced in the present research, coincides with an earlier study, where 93% of mortality from cancer in Colombia was well certified [38]. A Mexican analysis showed that some types of cancer present a better concordance [39]. The high concordance with COVID-19 diagnoses could be due to the standardized form that this new diagnosis was added to the list, as it includes information on the condition that started the causal chain of death [40].

The death certificate may have limitations as a source of mortality data when estimating deaths due to rare events because they are not certified. Underreporting of these events may underestimate the true burden of the disease [41]. The capacity of ICD-10 to track deaths from rare events is limited, so little is reported in mortality studies [42]. Furthermore, they are not recorded because proximal causes of death mask more distal causes of death, training to complete death certificates is insufficient, opinions on how comorbidities affect the final CoD are disparate [41], and lack of access to medical care. For the results of the study to be generalizable, the sample must be representative of the population; however, for rare events, it may be difficult to obtain a representative sample due to the low frequency of these events.

With few observations, the Kappa value should be interpreted with caution because inherent variability can affect its precision. With a small number of observations, it can provide valuable information on the agreement between measurement methods. Furthermore, its confidence interval allows us to get an idea of the variability associated with the result [43].

The discrepancies observed in CoD assignment can be contextualized within the broader challenges of death certification in Colombia. A substantial proportion of death certificates (79%) were prepared in health services by doctors, in some cases by health professionals with limited experience in this specific task or with a greater workload, which influences in not carefully reviewing the medical records to formulate the CoD sequence [29]. Regarding deaths at home, some lack complete data in the medical records, experiencing greater difficulty in determining CoDs, and relatives may not provide enough information. In rural areas, certification is carried out by other untrained health professionals, introducing possible errors in the certification process [30].

Certification disparities exist between the certification of violent and non-violent deaths. Non-violent deaths are certified by the treating physician or other health professionals authorized to certify them, and violent deaths are certified by the forensic doctor [31]. In terms of legal connotations of death certificates, it is true that the certifier must recognize the potential effects that the logical elaboration of the CoD will have. Using certain CoDs may provide information that could be used to challenge, reduce, or deny life insurance benefits. Additionally, the death certificate is rarely used as the sole source of information for legal or insurance purposes. Nevertheless, to reduce or avoid these situations, it is crucial to obtain correct and accurate information at all times [32].

This analysis has limitations. First, the determination of CoD was restricted to the information available in medical records, which may also be imperfect. Second, in deaths where the clinical documentation was incomplete or in the event of a death that occurred at home, relatives or acquaintances of the deceased were interviewed, presenting possible recall bias. Third, potential observer bias in the health professionals who analyzed the medical records, due to the subjectivity. However, the risk of bias is low because the health professionals who analyzed the medical records were trained and were under permanent supervision in the information collection process. Fourth, the use of broad categories at the ICD-10-chapter level could limit the identification of inaccuracies and the specificity of the analysis. To mitigate this situation, a more detailed analysis was included employing ICD-10 mortality tabulation list 2–80 causes plus COVID-19. This study had methodological strengths such as the use of a gold standard death certificate, completed by trained health professionals. Additionally, it was based on solid criteria for the selection of cases in multiple jurisdictions at the national level, relying on complete medical records or complemented with information given by relatives, representing a more precise analysis of the death certification process in Colombia. The differences between some of the CoDs identified between the original and gold standard death certificates are associated with the heterogeneity of the territories, where the death certification process varies depending on factors such as the lack of training of health professionals, incomplete medical records, difficulties in determining the CoDs, or internet access [15,22,31].

In conclusion, our study sheds light on the commendable quality of underlying CoD information in Colombian death certificates, reflecting a 74% concordance with the gold standard death certificate. On the other hand, the general kappa statistic by sex was high for the CoD of infections, digestive diseases, perinatal conditions, congenital anomalies and signs, symptoms, and poorly defined conditions; and almost perfect for neoplasms and COVID-19.

The accuracy and quality of the information provided by death certificates cannot be understated, as they serve as a valuable source of public health information [44]. The information they provide supports public health authorities in making decisions. In this sense, the continuing education of health personnel, adequate training in the university curriculum and in the workplace, formalized processes for reviewing death certifications, the use of automated systems to record the cause(s) of death and selecting the underlying CoD can improve accuracy in death certification [8,35,44,45].

To enhance the accuracy of the death certification process, we recommend robust educational and training initiatives for health professionals, the development of comprehensive guidelines for death certificate preparation, and ongoing monitoring in health services to identify and address information quality issue. These measures are imperative for refining mortality statistics and ensuring the reliability of CoD data in the Colombian context.

## Supporting information

**S1 File. Appendix telephone interview guide.**
(DOCX)

**S2 File. Database accuracy: we confirm that uploaded data files contain no data which could be used to identify study participants.**
(XLSX)

**S1 Table. Underlying CoD agreement metrics between original and gold standard death certificates according to ICD-10 mortality list 2. Colombia, 2021.**
(DOCX)

## Acknowledgments

Special thanks to Dr. Richard Garfield of the Centers for Disease Control and Prevention for his valuable technical support during the research development.

## Author contributions

**Conceptualization:** Pablo Chaparro-Narváez, Jessika Alexandra Manrique Sanchez, Laura Berrio-Parra, Diana Carolina Urrego Ricaurte, Luis José Torres-Rojas, Nidia Patricia Orjuela Cantor, Claudia Patricia Mora Aguirre, Yesid Rojas Quevedo, Clara Suárez, Carlos Castañeda-Orjuela.

**Data curation:** Pablo Chaparro-Narváez, Jessika Alexandra Manrique Sanchez, Laura Berrio-Parra, Diana Carolina Urrego Ricaurte, Luis José Torres-Rojas, Nidia Patricia Orjuela Cantor, Claudia Patricia Mora Aguirre, Yesid Rojas Quevedo, Clara Suárez.

**Formal analysis:** Pablo Chaparro-Narváez, Jessika Alexandra Manrique Sanchez, Laura Berrio-Parra, Diana Carolina Urrego Ricaurte, Luis José Torres-Rojas, Nidia Patricia Orjuela Cantor, Claudia Patricia Mora Aguirre, Yesid Rojas Quevedo, Clara Suárez, Carlos Castañeda-Orjuela.

**Funding acquisition:** Pablo Chaparro-Narváez, Carlos Castañeda-Orjuela.

**Investigation:** Pablo Chaparro-Narváez, Jessika Alexandra Manrique Sanchez, Laura Berrio-Parra, Diana Carolina Urrego Ricaurte, Luis José Torres-Rojas, Nidia Patricia Orjuela Cantor, Claudia Patricia Mora Aguirre, Yesid Rojas Quevedo, Clara Suárez, Carlos Castañeda-Orjuela.

**Methodology:** Pablo Chaparro-Narváez, Jessika Alexandra Manrique Sanchez, Laura Berrio-Parra, Diana Carolina Urrego Ricaurte, Luis José Torres-Rojas, Nidia Patricia Orjuela Cantor, Claudia Patricia Mora Aguirre, Yesid Rojas Quevedo, Clara Suárez, Carlos Castañeda-Orjuela.

**Project administration:** Pablo Chaparro-Narváez, Jessika Alexandra Manrique Sanchez, Carlos Castañeda-Orjuela.

**Resources:** Pablo Chaparro-Narváez.

**Software:** Pablo Chaparro-Narváez, Jessika Alexandra Manrique Sanchez, Laura Berrio-Parra, Diana Carolina Urrego Ricaurte, Luis José Torres-Rojas, Nidia Patricia Orjuela Cantor, Claudia Patricia Mora Aguirre, Yesid Rojas Quevedo.

**Supervision:** Pablo Chaparro-Narváez, Jessika Alexandra Manrique Sanchez, Carlos Castañeda-Orjuela.

**Validation:** Pablo Chaparro-Narváez, Jessika Alexandra Manrique Sanchez, Laura Berrio-Parra, Diana Carolina Urrego Ricaurte, Luis José Torres-Rojas, Nidia Patricia Orjuela Cantor, Claudia Patricia Mora Aguirre, Yesid Rojas Quevedo, Carlos Castañeda-Orjuela.

**Visualization:** Pablo Chaparro-Narváez, Jessika Alexandra Manrique Sanchez, Laura Berrio-Parra, Diana Carolina Urrego Ricaurte, Luis José Torres-Rojas, Nidia Patricia Orjuela Cantor.

**Writing – original draft:** Pablo Chaparro-Narváez, Jessika Alexandra Manrique Sanchez, Laura Berrio-Parra, Diana Carolina Urrego Ricaurte, Luis José Torres-Rojas, Nidia Patricia Orjuela Cantor, Claudia Patricia Mora Aguirre, Yesid Rojas Quevedo, Clara Suárez, Carlos Castañeda-Orjuela.

**Writing – review & editing:** Pablo Chaparro-Narváez, Jessika Alexandra Manrique Sanchez, Laura Berrio-Parra, Diana Carolina Urrego Ricaurte, Luis José Torres-Rojas, Nidia Patricia Orjuela Cantor, Carlos Castañeda-Orjuela.

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
