## [Decision Letter · Decision Letter 0]

1 Dec 2023

PONE-D-23-33548Validity of the information on the basic cause of death: an analysis in Colombia during the COVID-19 pandemic in 2021PLOS ONE

Dear Dr. Berrio-Parra,

Thank you for submitting your manuscript to PLOS ONE. After careful consideration, we feel that it has merit but does not fully meet PLOS ONE’s publication criteria as it currently stands. Therefore, we invite you to submit a revised version of the manuscript that addresses the points raised during the review process.

Your article addresses a crucial issue concerning the disparities between causes of death (CoD) in death certificates and other sources including clinical records, which form the basis for much epidemiological and demographical data, influencing healthcare policy and decision-making in Colombia and beyond. While the study's objective is clear, the manuscript lacks sufficient detail to meet PLOS ONE's publication criteria. I am outlining the main issues that must be addressed for the manuscript to be considered suitable for publication in our journal.

1. Clear definition of PICO: Please restate the PICO structure research question clearly, focusing on the "reference gold standard CoD source" and the "original CoD source." Provide detailed selection criteria, sampling methods, and information about the study population. Revise all calculations thoroughly to ensure accurate comparisons, and report the main outcomes according to the planned methods to prevent spin bias.

2. Reproducible methods and research integrity compliance: To enhance reproducibility, provide full details about the CoD adjudicating committee and methods for determining causes of death.

3. Conclusions supported by data: Ensure that the discussion section addresses the implications of your research findings. Most importantly, suggest possible solutions that could improve accurate CoD adjudication, serving as a foundation for better healthcare policy and decision-making.

4. Reporting guidelines and community standards: Comply with the IMRaD structure for scientific articles, including quality content items for the abstract section. It is highly recommended to follow the abstract and full-text STROBE statements for observational studies.

5. Article is written in an intelligible fashion and is in standard English: Perform a thorough English copyediting and coherence revision to ensure clear understanding of the research. Present manuscript sections, tables, and figures in an organized and appropriately classified manner.

We look forward to receiving your revised manuscript.

Kind regards,

Andres Mauricio Acevedo-Melo, M.D.

Academic Editor

PLOS ONE

Journal Requirements:

Reviewers' comments:

Reviewer's Responses to Questions

**Comments to the Author**

1. Is the manuscript technically sound, and do the data support the conclusions?

Reviewer #1: Partly

Reviewer #2: No

Reviewer #3: Partly

2. Has the statistical analysis been performed appropriately and rigorously? 

Reviewer #1: Yes

Reviewer #2: Yes

Reviewer #3: Yes

3. Have the authors made all data underlying the findings in their manuscript fully available?

Reviewer #1: No

Reviewer #2: Yes

Reviewer #3: Yes

4. Is the manuscript presented in an intelligible fashion and written in standard English?

Reviewer #1: Yes

Reviewer #2: Yes

Reviewer #3: Yes

5. Review Comments to the Author

Reviewer #1: The manuscript presents several methodological concerns, including unclear processes for informed consent when accessing clinical records, potential legal ramifications due to the sensitive nature of death diagnoses, undisclosed qualifications of the data reviewers, and doubts about data anonymization and interview consent.

Reviewer #2: Although the document presents a well-known statistical technique, such as Kappa, it is nothing more than an academic exercise that in real life does not advance the possibility of correcting poorly coded deaths for better decision making.

Reviewer #3: Dear authors, the paper is interesting and analyzes a crucial problem. Nevertheless, I suggest you an additional effort to better clarify some aspects of the methods and to improve the consistency of data analysis.

General remarks

1. The declared objective of the paper (line 82-85) is to compare the ‘original certificate’ with a more accurate information, so the “standard certificate” should be always used as the gold standard in the statistical analysis. Nevertheless, some indicators are calculated taking the original certificate as a reference. For instance, given a cause of death J, “false negatives” are certificates classified as J with the gold standard and classified differently by the “original certificate” (which is under evaluation). “False positives” should be certificates in which the cause is other than J for the gold standard J for the “original certificate”. For example, for ICD10 chapter I, the false positives are 9 (from table 1), and the percentage of false positive 1.20. In the same way, “false negatives” are 8 and the percentage of false negative is 8/23*100= (34.8). (see table in the attached file). Nevertheless, in table 3 the percentages of false positives and false negatives are calculated using the original certificate as the gold standard. I suggest the authors to calculate the indicators using the standard certificate as the reference.

The indicator presented in line 114 seems to be calculated consistently with the objectives (standard certificate as gold standard) but the results presented in the last column of table 3 seem to have inverted signs.

In the discussion, line 207, the authors report that there is an underestimation of (for instance) circulatory system disease, but in reality, the original certificate overestimates these causes compared to the gold standard statistics.

What the authors think about this? It would be better to calculate the indicators using the standard certificate as gold standard.

2. I suggest to clarify the methods (Ln 95-106):

a. The trained health professionals, fill-in a new death form (part 1 and part 2 of the standard death certificate) or attribute a single cause-of-death?

b. Does the underlying cause codes for the original certificate correspond to those of official statistics for Colombia?

c. Is there a hierarchy of the medical documentation used for attributing the cause of death for the standard certificate? For deaths with more than one type of medical document (necropsy, clinical records, interview to relatives etc) only one is used for attributing the cause of death (as it appears from table 1)? In which order?

Additional specific topics:

1. I suggest to use the standard term “underlying cause” instead of “basic cause”.

2. When ICD chapters are mentioned, I would suggest to use also the chapter number, that would allow to find the results more easily in the tables.

3. Acronyms should be explained only once, the first time that are used (examples in line 28, CoD acronym described in the line above).

4. Ln 32-33. The sentence is not clear, what the author mean by “chart review” and “those who knew the medical history”? The terms are clear in the paper but they should be clear also in the abstract. Should it be “medical documentation” and “interviews to relatives”? Ln 33: the certificate considered “standard” is the one based on the medical documentation and interviews?

5. Ln 116-117, which variable and which points are plotted in the Bland-Altman diagram?

6. Ln 124-130 and table 1. Is the sample analyzed (776 cases) representative of the total deaths in Colombia (in terms of age and sex)? A sentence regarding this aspect and on the possible distortion due to the loss of some cases from the desired sample would be useful.

7. Ln 157-164. Confidence interval for Kappa should be used for comments and only robust values should be highlighted. For instance, results found for chapter “pregnancy childbirth and puerperium” are based only on two cases: it is not possible to evaluate false positives and Kappa is misleading. This should be taken into account.

8. Ln 167: The term “level of occurrence” is not clear. The table represents the Deaths by underlying cause according to the standard and original certificate and comparison of figures deriving from the two certificates.

9. Table 3. Last column (Change in cause – specific mortality fraction (%)): if the number of deaths for the original certificate is greater than for the standard, why the sign of the indicator is negative? According to the formula (line 114) I would expect a positive sign (and vice-versa for positive values).

10. Ln 171-177: This part presents the results for men and women (is it correct?), but it seems a repetition of the above comment. I suggest to mention that for most causes the results are comparable in the two genders and successively to focus only on chapters where there are differences.

11. Ln 177: Fig 1 is a table, it should be considered table 4.

12. Ln 182: The term validity in this context is not clear (the description of this table should be similar to table 2 which has similar content).

13. Figure 1 (should be table 4): the columns’ titles for women are different than for men and they have different order.

14. Ln 207-209: The authors report underestimation for ‘circulatory system disease’ but the results seem to show that the original death certificate overestimates them. The same occurs for ‘diseases of respiratory system’. This term should be used when the original death certificate, which is under evaluation, detects more cases than the standard. The use of the term underestimation considering the false negatives is misleading. I would suggest to use the CSMF, which is calculated but little used in comments.

15. Ln 210: The result about ‘pregnancy, childbirth and puerperium’ is very counterintuitive, since death certificates are often considered to underestimate maternal mortality, while examining additional documentation should allow to detect more accurately the cases. What is the authors’ opinion on this result? In addition, it is surprising that there are 7 cases classified in chapter XVII (Symptoms and signs) by the standard certificates which is based on additional documentation. Is there some explanation for these cases?

16. Ln 216-218: Could authors rephrase this sentence? It is not very clear.

17. Ln 224: Are there differences for deaths occurred in hospital compared to other places? Are there differences for source of determination of the cause of death? If possible, some comments could be provided in the discussion (even if not provided in the results).

6. PLOS authors have the option to publish the peer review history of their article (what does this mean? ). If published, this will include your full peer review and any attached files.

**Do you want your identity to be public for this peer review?** For information about this choice, including consent withdrawal, please see our Privacy Policy .

Reviewer #1: No

Reviewer #2: No

Reviewer #3: **Yes: ** Francesco Grippo

---

## [Author Response · Author response to Decision Letter 1]

26 Feb 2024

We gratefully received the revisor comments on our manuscript. Therefore, we made several adjustments to the manuscript addressing all aspects for its publication. In the response letter to the reviewers, we sent all the issues in which there were doubts, comments or suggestions. We uploaded a response letter to the reviewers, the revised manuscript with tracked changes, the manuscript, a figure, ethics committee approval, and supporting information files with all relevant data..We hope that the adjustments have responded to the questions and comments sent by the reviewers and that the uploaded files are in accordance with the requirements of Plos One. We are awaiting for the new revision!

---

## [Decision Letter · Decision Letter 1]

9 Apr 2024

PONE-D-23-33548R1Accuracy of Information on the Underlying Cause of Death: An Analysis in Colombia during the COVID-19 Pandemic in 2021PLOS ONE

Dear Dr. Berrio-Parra,

Thank you for submitting your manuscript to PLOS ONE. After careful consideration, we feel that it has merit but does not fully meet PLOS ONE’s publication criteria as it currently stands. Therefore, we invite you to submit a revised version of the manuscript that addresses the points raised during the review process. Hereby I provide the main concerns:

*Originality and Context: Please provide a clearer rationale for the study's significance within the existing literature.

*Methodological Clarity and Ethical Considerations: There are several methodological and ethical considerations that need to be addressed. The temporal scope of mortality records needs clarification, along with details on informed consent procedures, mitigation of biases during data review, and anonymization processes. Please ensure transparency and provide clear explanations for these aspects in the Methods section.

*Engagement with Data Sources: Clarify the decision to use data from the National Institute of Health (INS) instead of the National Administrative Department of Statistics (DANE).

*Statistical Analysis and Results: Provide a more detailed explanation of the statistical analysis, including the calculation of false positives and negatives and the rationale behind using kappa statistics solely for establishing validity. Correct any errors in the presentation of results, such as those regarding circulatory diseases.

*Representativeness and Underestimation: Substantiate claims about underestimation of causes of death, especially for rare events, and clarify the representativeness within the sample.

*Grammar and style review: Please perform a thorough revision of the manuscript to meet PLOS ONE Criteria for Publication.

We look forward to receiving your revised manuscript.

Kind regards,

Andres Mauricio Acevedo-Melo, M.D.

Academic Editor

PLOS ONE

Reviewers' comments:

Reviewer's Responses to Questions

**Comments to the Author**

1. If the authors have adequately addressed your comments raised in a previous round of review and you feel that this manuscript is now acceptable for publication, you may indicate that here to bypass the “Comments to the Author” section, enter your conflict of interest statement in the “Confidential to Editor” section, and submit your "Accept" recommendation.

Reviewer #1: All comments have been addressed

Reviewer #4: (No Response)

2. Is the manuscript technically sound, and do the data support the conclusions?

Reviewer #1: Partly

Reviewer #4: Partly

3. Has the statistical analysis been performed appropriately and rigorously? 

Reviewer #1: (No Response)

Reviewer #4: Yes

4. Have the authors made all data underlying the findings in their manuscript fully available?

Reviewer #1: (No Response)

Reviewer #4: Yes

5. Is the manuscript presented in an intelligible fashion and written in standard English?

Reviewer #1: (No Response)

Reviewer #4: No

6. Review Comments to the Author

Reviewer #1: (No Response)

Reviewer #4: The paper is interesting but a paragraph in the Methods section is needed providing the context of the study, specifically addressing the following points of death certification in Colombia:

1) Who fills the death certificates (this is already reported in Discussion, but a brief anticipation is needed in Methods)

2) Which type of death certificate is adopted: exactly the International Death Certificate from the WHO, or an adaption?

3) In national mortality statistics, how the underlying cause of death is selected: by trained coders, or by a software as in the US (ACME) or in Europe (IRIS)?

4) Which version of the ICD-10 was adopted in the study period?

Moreover, in Methods authors should clearly specify the following:

1) What are they comparing, e.g. the disease on the underlying cause line in the death certificate compiled by the certifier, or the underlying cause of death selected according international coding rules by a coder or by software? This might not be important for the gold standard, since in a “perfect” death certificate the two conditions should coincide, but is relevant for the original death certificate: as an example, in a death certificate reporting pulmonary embolism in part I and cancer in part II, the latter (and not pulmonary embolism) will be selected according to international coding rules as the CoD

2) Authors should explicitly state already in Methods that they are measuring agreement at the chapter level of the ICD-10

Results need some improvement: a footnotes in all Tables with an abbreviated description of chapter numbers (e.g. “Infectious” for chapter I). Moreover, in page 11, last lines, an error was left from the previous version (circulatory diseases exhibited higher percentages of false positives, not of false negatives)

Lastly, authors should carefully check for typos through the manuscript: only as a first example, at the end of Methods in the Abstract, “…. and calculate kappa value were utilized” should be replaced with “… and kappa value were utilized …..”

7. PLOS authors have the option to publish the peer review history of their article (what does this mean? ). If published, this will include your full peer review and any attached files.

**Do you want your identity to be public for this peer review?** For information about this choice, including consent withdrawal, please see our Privacy Policy .

Reviewer #1: No

Reviewer #4: No

---

## [Author Response · Author response to Decision Letter 2]

12 Jul 2024

We gratefully received the reviewr´s comments on our manuscript. Therefore, we made the other adjustments to the manuscript addressing all aspects for its publication. We uploaded a response letter to the reviewers, the revised manuscript with tracked changes, the manuscript, a figure, ethics committee approval, appendix telephone interview guide and supporting information files with all relevant data.We hope that the adjustments have responded to the questions and comments sent by reviewers and that the uploaded files are in accordance with the requirements of Plos One. We will waiting for the new review.

---

## [Decision Letter · Decision Letter 2]

5 Aug 2024

PONE-D-23-33548R2Accuracy of Information on the Underlying Cause of Death: An Analysis in Colombia during the COVID-19 Pandemic in 2021PLOS ONE

Dear Dr. Berrio-Parra,

Thank you for submitting your manuscript to PLOS ONE. After careful consideration, we feel that it has merit but does not fully meet PLOS ONE’s publication criteria as it currently stands. Therefore, we invite you to submit a revised version of the manuscript that addresses the points raised during the review process.

We look forward to receiving your revised manuscript.

Kind regards,

Andres Mauricio Acevedo-Melo, M.D.

Academic Editor

PLOS ONE

Journal Requirements:

Additional Editor Comments:

Dear Authors,

Thank you for submitting a revised version of your manuscript. Before it can be considered for publication, there are a few issues that need to be addressed. Please review the following main points, which are related to PLOS ONE publication criteria and highlight the concerns raised by the reviewers:

*Conclusions Supported by Data: Your manuscript uses the terms “accuracy” and “agreement” interchangeably. However, it is important to note that kappa statistic measures agreement, while false positive and negative rates estimate discrimination. Additionally, the distinction between “original” and “standard” death certificates, role of participating institutions and custodianship need clarification. Please revise these concepts throughout the manuscript to ensure clarity. Also, provide clear descriptions and interpretations of measures such as the Change in Cause-Specific Mortality Fraction (CCSMF) and the Bland-Altman diagram axis and legends.

*Clarity and Language: The manuscript should be written in clear, standard English. Please conduct a thorough English language review and address the mistakes highlighted by the reviewers. The following typos and issues were also noted: the word “Charts” is used in the abstract submission, while “records” is used in the manuscript abstract; a repeated sentence appears on lines 35-38; “offres” should be corrected to “offers” on line 56; and the acronym “IRIS” on line 112 should be explained. Please also review and correct subject-verb agreement and coherence issues, particularly between lines 280 and 284.

Reviewers' comments:

Reviewer's Responses to Questions

**Comments to the Author**

1. If the authors have adequately addressed your comments raised in a previous round of review and you feel that this manuscript is now acceptable for publication, you may indicate that here to bypass the “Comments to the Author” section, enter your conflict of interest statement in the “Confidential to Editor” section, and submit your "Accept" recommendation.

Reviewer #1: All comments have been addressed

Reviewer #4: All comments have been addressed

2. Is the manuscript technically sound, and do the data support the conclusions?

Reviewer #1: No

Reviewer #4: Yes

3. Has the statistical analysis been performed appropriately and rigorously? 

Reviewer #1: No

Reviewer #4: Yes

4. Have the authors made all data underlying the findings in their manuscript fully available?

Reviewer #1: Yes

Reviewer #4: Yes

5. Is the manuscript presented in an intelligible fashion and written in standard English?

Reviewer #1: Yes

Reviewer #4: Yes

6. Review Comments to the Author

Reviewer #1: The manuscript needs clarity on data sources, terminology, and the impact of the COVID-19 pandemic on death certificate accuracy.

Reviewer #4: Author improved the manuscript. Some minor revision is still needed

1) Abstract: please remove chapter numbers ("circulatory system diseases" is enough for the Abstract, without adding "Chapter IX")

2) Careful editing is needed, only as an example Introduction, page 3, row 56: please change "offres" with "offers"

3) Please correct Table 1: under "Reviewer underlying cause", change the right column from "Diagnosis X" to "The others (or better, "All other diagnoses"), and check all the following formulas

7. PLOS authors have the option to publish the peer review history of their article (what does this mean? ). If published, this will include your full peer review and any attached files.

**Do you want your identity to be public for this peer review?** For information about this choice, including consent withdrawal, please see our Privacy Policy .

Reviewer #1: No

Reviewer #4: No

---

## [Author Response · Author response to Decision Letter 3]

17 Sep 2024

We gratefully received the journal's and reviewer´s comments on our manuscript. Therefore, we made the other adjustments to the manuscript addressing all aspects for its publication. We uploaded the response letter to the reviewers, the revised manuscript with tracked changes, the manuscript, a figure, ethics committee approval, appendix telephone interview guide and supporting information files with all relevant data.We hope that the adjustments have responded to the questions and comments sent by reviewers and that the uploaded files are in accordance with the requirements of Plos One. We will waiting for the new review.

---

## [Decision Letter · Decision Letter 3]

5 Nov 2024

PONE-D-23-33548R3Agreement of information on the underlying cause of death: an analysis in Colombia during the COVID-19 pandemic in 2021PLOS ONE

Dear Dr. Berrio-Parra,

Thank you for submitting your manuscript to PLOS ONE. After careful consideration, we feel that it has merit but does not fully meet PLOS ONE’s publication criteria as it currently stands. Therefore, we invite you to submit a revised version of the manuscript that addresses the points raised during the review process.

We look forward to receiving your revised manuscript.

Kind regards,

Pasyodun Koralage Buddhika Mahesh

Academic Editor

PLOS ONE

Journal Requirements:

Reviewers' comments:

Reviewer's Responses to Questions

**Comments to the Author**

1. If the authors have adequately addressed your comments raised in a previous round of review and you feel that this manuscript is now acceptable for publication, you may indicate that here to bypass the “Comments to the Author” section, enter your conflict of interest statement in the “Confidential to Editor” section, and submit your "Accept" recommendation.

Reviewer #4: All comments have been addressed

Reviewer #5: (No Response)

Reviewer #6: All comments have been addressed

2. Is the manuscript technically sound, and do the data support the conclusions?

Reviewer #4: Yes

Reviewer #5: Yes

Reviewer #6: Yes

3. Has the statistical analysis been performed appropriately and rigorously? 

Reviewer #4: Yes

Reviewer #5: Yes

Reviewer #6: Yes

4. Have the authors made all data underlying the findings in their manuscript fully available?

Reviewer #4: Yes

Reviewer #5: Yes

Reviewer #6: Yes

5. Is the manuscript presented in an intelligible fashion and written in standard English?

Reviewer #4: Yes

Reviewer #5: Yes

Reviewer #6: Yes

6. Review Comments to the Author

Reviewer #4: Authors addressed my previous comments

Reviewer #5: The document is a manuscript draft discussing the agreement of information on the underlying cause of death in Colombia during the COVID-19 pandemic in 2021.

The key findings of the study by the authors are:

The agreement between the original and gold standard death certificates, categorized according to ICD-10 chapters, was 74%. Higher levels of agreement were observed for "codes for special COVID-19 situations" (kappa=0.84) and neoplasms (kappa=0.84). COVID-19 was considered the underlying cause of death in 29.8% of the original certificates and 31.8% of the gold standard certificates, with a very good agreement (kappa=0.84). The proportion of agreement for most causes of death remained consistent between genders, with 73% agreement in men and 75% in women.

The study recommends robust educational and training initiatives, the development of comprehensive guidelines, ongoing monitoring, the use of automated systems, and formalised review processes to improve the death certification process in Colombia.

I congratulate the authors for selecting this all-important topic and conducting the study to identify potential misclassifications in the cause of death data. Using Iris automated coding software for mortality coding has ensured the correct and consistent application of coding rules to select the underlying cause of death.

My primary concern is selecting the broad categories (ICD Chapter level) to compare the original UCOD with the gold standard. Grouping causes of death at the ICD chapter level can be considered a broad categorisation, which may limit the granularity and specificity of the analysis. Here are some points to consider:

1. ICD chapters encompass a wide range of diseases and conditions. For example, the chapter on "Diseases of the circulatory system" includes various conditions such as myocardial infarction, strokes, and hypertension. Grouping at this level may mask discrepancies in specific causes of death within the chapter.

2. More detailed categorization, such as using ICD-10 codes at the sub-chapter or individual code level, could provide a clearer picture of the accuracy and quality of cause of death data. It would allow for identifying specific areas where misclassification or inaccuracies are more prevalent. The authors could easily use the mortality tabulation lists (General mortality 1 -103 causes or General mortality 2 – 80 causes) provided in ICD-10 for disease categorisation.

3. Detailed cause of death data is crucial for designing targeted public health interventions. Broad categories may not provide the necessary details to inform specific health policies and resource allocation.

4. The study acknowledges the limitations of using broad categories and suggests that a more detailed analysis could provide additional insights. However, the broad categorisation might have been chosen for practical reasons, such as data availability or analysis feasibility.

5. Future studies could benefit from a more detailed examination of cause of death data using specific ICD-10 codes. This would enhance the understanding of the quality of death certification and help identify specific areas for improvement.

In summary, while the broad categorization at the ICD chapter level provides a general overview of the agreement between original and gold standard death certificates, a more detailed analysis would offer greater specificity and potentially more actionable insights.

Another issue is that he chapter-level broad categories of causes of death could include garbage codes (unusable codes for public health decision-making) of medium impact. These are known as ‘Level 3 (medium)’ garbage codes, meaning codes with important implications: causes for which the true underlying cause of death is likely to be within the same ICD chapter. For instance, ‘unspecified cancer’ still provides enough information to know if the UCOD was due to cancer. However, knowledge about the site of cancer is important for public health policy since different strategies are applied for different types (sites) of cancer (i.e. breast versus lung cancer).

The categorization of garbage codes into levels, particularly Level 3 garbage codes, is a useful approach to understanding the quality of cause of death data. Level 3 garbage codes refer to causes of death where the actual underlying cause is likely within the same ICD chapter.

In summary, since the study's use of broad categories aligns with the concept of Level 3 garbage codes, it still needs to be revised in terms of specificity and potential misclassification. Recognizing and addressing the presence of Level 3 garbage codes can help improve the quality of cause-of-death data and enhance the utility of mortality statistics for public health.

While the study methodology is generally sound, there are a few potential issues and limitations that could be addressed:

1. As previously mentioned, grouping causes of death at the ICD chapter level is quite broad and may mask specific discrepancies within those chapters. More granular categorization could provide a clearer picture of the accuracy and quality of cause of death data.

2. Determining the gold standard cause of death relied heavily on medical records and, when necessary, interviews with relatives or witnesses. Medical records may not always be complete or accurate, and interviews can introduce recall bias.

3. Although the health professionals were trained and supervised, there is still a risk of observer bias in interpreting medical records and assigning causes of death.

4. The study used data from the DANE 2021 mortality database and medical records. It did not include other potential sources of information, such as autopsy reports, which could provide additional accuracy in determining the cause of death.

5. The study focused on 92 municipalities, which may only partially represent part of the country. Geographical and demographic variability in the quality of death certification could exist, but this is not captured in this sample.

6. The study was conducted during the COVID-19 pandemic, which could have affected the accuracy of death certification due to the high volume of deaths and the strain on healthcare systems. This context should be considered when interpreting the results.

7. While the study received ethical approval and took measures to anonymise data, obtaining consent from relatives or witnesses for interviews could introduce ethical concerns, especially in sensitive or traumatic deaths.

8. The findings may not be generalizable to other countries or regions with different healthcare systems, death certification processes, and levels of training for health professionals.

9. Although the study mentions that health professionals were trained and supervised, it needs to provide detailed information on the extent and quality of this training. Inadequate training could affect the reliability of the gold standard in determining the cause of death.

10. While the sample size is substantial, the study does not discuss potential selection bias in the choice of municipalities and deaths included in the sample. Selection bias could affect the representativeness of the findings.

Addressing these issues in future research could enhance the robustness and reliability of the findings and provide more detailed insights into the quality of cause of death data.

Reviewer #6: This article aims to estimate the agreement of the underlying Cause of Death in the original death certificate, compared with a gold standard certificate based on information from medical records and relatives, in Colombia during 2021. The authors have used summary measures of agreement such as the proportion of agreement, Kappa, CCSMF and graphical methods such as the Bland-Altman plot. While commending their efforts, the following comments are given with the hope that these will benefit the authors.

1. Please elaborate on the two-stage sampling method a bit more. As an example, it is mentioned that in the second stage, 92 municipalities were selected. It would be better to add that these 92 were selected out of how many. Also, what method was used in this random sampling (i.e. whether done by computer-generated random numbers etc.)? Furthermore, it is mentioned that 33 territorial entities were “considered” in the first stage. It would be nicer to elaborate a bit more on how this consideration was done.

2. Please give a reference to the sample size calculation formula/ software that was used.

3. In the sample size calculation, please define what variable was taken as “expected proportion of 70%” (i.e. whether it is the expected proportion of agreement?). Was this figure selected based on some previous literature or by some other method?

4. Please add any rationale for choosing the design effect of 2.5

5. It is mentioned that “In cases where medical 143 records lacked sufficient information, interviews with relatives, acquaintances, or witnesses 144 were conducted". Please add a bit more details on who conducted these interviews, in what settings and what kind of information were asked from the respondent etc.

7. PLOS authors have the option to publish the peer review history of their article (what does this mean? ). If published, this will include your full peer review and any attached files.

**Do you want your identity to be public for this peer review?** For information about this choice, including consent withdrawal, please see our Privacy Policy .

Reviewer #4: No

Reviewer #5: No

Reviewer #6: **Yes: ** I.O.K.K.Nanayakkara

---

## [Author Response · Author response to Decision Letter 4]

20 Jan 2025

We gratefully received the journal's and reviewer´s comments on our manuscript. Therefore, we made the adjustments to the manuscript addressing all aspects for its publication. We uploaded the response letter to the reviewers, the revised manuscript with tracked changes, the manuscript, a figure, a supplementary table, ethics committee approval, appendix telephone interview guide and supporting information files with all relevant data.We hope that the adjustments have responded to the questions and comments sent by reviewers and that the uploaded files are in accordance with the requirements of Plos One. We will waiting for the new review.

---

## [Decision Letter · Decision Letter 4]

19 Feb 2025

Agreement of information on the underlying cause of death: an analysis in Colombia during the COVID-19 pandemic in 2021

PONE-D-23-33548R4

Dear Dr. Berrio-Parra,

We’re pleased to inform you that your manuscript has been judged scientifically suitable for publication and will be formally accepted for publication once it meets all outstanding technical requirements.

Kind regards,

Pasyodun Koralage Buddhika Mahesh

Academic Editor

PLOS ONE

Additional Editor Comments (optional):

Reviewers' comments:

Reviewer's Responses to Questions

**Comments to the Author**

1. If the authors have adequately addressed your comments raised in a previous round of review and you feel that this manuscript is now acceptable for publication, you may indicate that here to bypass the “Comments to the Author” section, enter your conflict of interest statement in the “Confidential to Editor” section, and submit your "Accept" recommendation.

Reviewer #5: All comments have been addressed

Reviewer #6: All comments have been addressed

2. Is the manuscript technically sound, and do the data support the conclusions?

Reviewer #5: Yes

Reviewer #6: Yes

3. Has the statistical analysis been performed appropriately and rigorously? 

Reviewer #5: Yes

Reviewer #6: Yes

4. Have the authors made all data underlying the findings in their manuscript fully available?

Reviewer #5: Yes

Reviewer #6: Yes

5. Is the manuscript presented in an intelligible fashion and written in standard English?

Reviewer #5: Yes

Reviewer #6: Yes

6. Review Comments to the Author

Reviewer #5: Thank you very much for your responses. It is great to see that you have responded taken appropriate action to all my comments. They have provided detailed responses and made necessary adjustments to the manuscript to address the concerns raised. Here are the key points of how you addressed the comments:

1. ICD Chapters and Detailed Categorization

The authors acknowledged the broad categorisation issue and incorporated the ICD-10 mortality tabulation list 2 (80 causes) for a more detailed analysis. They included the results in the supplement (Table 1S) and discussed the limitations of broad categories in the manuscript.

2. Granular Categorization

They agreed with the need for more detailed categorization and implemented it in the revised version. In the discussion section, they also discussed the impact of using broad categories and the benefits of detailed analysis.

3. Public Health Interventions

The authors concurred with the importance of detailed cause-of-death data for public health interventions and acknowledged the limitations of broad categories. They emphasized the need for future studies to use more detailed categorisation.

4. Garbage Codes

The authors recognized the importance of garbage codes and mentioned that Level 3 garbage codes were found in 2.84% of the basic CoD certifications, with most being "unspecified cancer."

5. Gold Standard and Observer Bias

They addressed the potential limitations of using medical records and interviews for the gold standard and acknowledged the risk of observer bias. They included these points in the discussion section as limitations.

6. Autopsy Reports

The authors clarified that autopsies are rarely performed for natural causes in Colombia, which is why they were not included in the study.

7. Geographical and Demographic Variability

They explained the probability sampling method used to ensure representativeness and provided additional details on the sampling process.

8. COVID-19 Pandemic Context

The authors included the context of the COVID-19 pandemic in the discussion, acknowledging its potential impact on the accuracy of death certification.

9. Ethical Concerns

They noted that no ethical concerns arose during the data collection process and emphasised the measures taken to ensure ethical standards.

10. Training of Health Professionals

The authors provided more details on the training given to health professionals by a DANE vital statistics expert.

11. Selection Bias

They explained the random sampling method used to minimise selection bias and provided details on the sampling process.

Overall, you have made comprehensive revisions to address my comments, enhancing the robustness and reliability of their findings. Therefore, I recommend that this manuscript be suitable for publication in your journal. Congratulations!

Reviewer #6: Thank you for submitting the revised manuscript and for addressing the reviewer’s comments thoughtfully. After carefully reviewing the changes, it is clear that the authors have made significant improvements, especially in clarifying the methodology and refining the discussion of the results.

The study is important, and the results contribute valuable insights to the understanding of cause of death classification in Colombia.

7. PLOS authors have the option to publish the peer review history of their article (what does this mean? ). If published, this will include your full peer review and any attached files.

**Do you want your identity to be public for this peer review?** For information about this choice, including consent withdrawal, please see our Privacy Policy .

Reviewer #5: **Yes: ** U S H GAMAGE

Reviewer #6: **Yes: ** I.O.K.K.Nanayakkara

---

## [Editor Report · Acceptance letter]

PONE-D-23-33548R4

PLOS ONE

Dear Dr. Berrio-Parra,

I'm pleased to inform you that your manuscript has been deemed suitable for publication in PLOS ONE. Congratulations! Your manuscript is now being handed over to our production team.

Kind regards,

on behalf of

Dr. Pasyodun Koralage Buddhika Mahesh

Academic Editor

PLOS ONE